# Deposition of Thick SiO_2_ Coatings to Carbonyl Iron Microparticles for Thermal Stability and Microwave Performance

**DOI:** 10.3390/s23031727

**Published:** 2023-02-03

**Authors:** Arthur V. Dolmatov, Sergey S. Maklakov, Anastasia V. Artemova, Dmitry A. Petrov, Artem O. Shiryaev, Andrey N. Lagarkov

**Affiliations:** Institute for Theoretical and Applied Electromagnetics RAS, Izhorskaya St. 13, 125412 Moscow, Russia

**Keywords:** core-shell particles, microwave measurements, absorbing materials, protective coatings

## Abstract

Thick dielectric SiO_2_ shells on the surface of iron particles enhance the thermal and electrodynamic parameters of the iron. A technique to deposit thick, 500-nm, SiO_2_ shell to the surface of carbonyl iron (CI) particles was developed. The method consists of repeated deposition of SiO_2_ particles with air drying between iterations. This method allows to obtain thick dielectric shells up to 475 nm on individual CI particles. The paper shows that a thick SiO_2_ protective layer reduces the permittivity of the ‘Fe-SiO_2_—paraffin’ composite in accordance with the Maxwell Garnett medium theory. The protective shell increases the thermal stability of iron, when heated in air, by shifting the transition temperature to the higher oxide. The particle size, the thickness of the SiO_2_ shells, and the elemental analysis of the samples were studied using a scanning electron microscope. A coaxial waveguide and the Nicholson–Ross technique were used to measure microwave permeability and permittivity of the samples. A vibrating-sample magnetometer (VSM) was used to measure the magnetostatic data. A synchronous thermal analysis was applied to measure the thermal stability of the coated iron particles. The developed samples can be applied for electromagnetic compatibility problems, as well as the active material for various types of sensors.

## 1. Introduction

Electromagnetic interference (EMI) is a challenge in electronic devices, radar, and antenna systems [1]. EMI is an unwanted electromagnetic radiation that acts as noise for electronic devices, as well as adversely affects human health [2,3,4]. This noise can originate from natural phenomena, e.g., thunder, solar flare, and electrostatic discharge [5], or can be caused by electronic devices. Nowadays, it is critical to develop effective materials capable of reducing the effects of EMI, i.e., electromagnetic shielding (EMI shielding), since all modern electronic devices emit electromagnetic interference.

In the past few years, magnetic nano- and microstructures have been in high demand in the development of materials that are effective absorbers in the microwave range. The simplicity and low cost of synthesis [6], along with high chemical stability and, at the same time, good biodegradability and biocompatibility [7], have made iron a preferred material, when compared with other transition metals in potential applications [8]. Iron is one of the most promising candidates for such applications as catalysis, microwave absorption, water purification, magnetic media materials, and many others. The most common forms of iron are oxides and hydroxides of iron, and carbonyl iron [8]. The production of iron nanoparticles is a complex technological challenge; ball mills, direct current arc plasma, and magnetic deposition methods are used to obtain such particles [9].

Of considerable interest is carbonyl iron (CI). Carbonyl iron possesses a high saturation magnetization [10], a large value of magnetic losses, along with a low permittivity of composites filled with CI powders. The magnetic properties of carbonyl iron can be changed by modification of the shape, size, or morphology of the iron [11]. For example, planar anisotropy, as observed in CI nanoflakes, improves the Snoek’s limit that increases both permeability and resonance frequency [12]. Such flake-type structures can neglect the skin effect in a high frequency range. Despite the fact that pure iron has a number of advantages, a high weight, processing difficulties, poor mechanical flexibility, and a narrow absorption bandwidth prevent their use in many engineering and shielding applications.

Certain optimization of the ratio between the dielectric and magnetic permeability of materials is required to better absorb EM energy due to magnetic and dielectric losses within the material. The main goal of present-day research is to create such functional materials, the electrodynamic characteristics of which can be varied with changing conditions for the synthesis of these materials, and, therefore, be optimized for certain applied problems [13]. With this in mind, researchers have proposed various strategies for creating EMI shielding materials and magnetic absorbers—one strategy is to create core-shell structures.

Core-shell structures are a new type of functional material; they have found applications in many fields, such as electronics [14], biomedicine [15], energy [14], optics [5], etc. The core and shell are usually made of different materials, the properties of the resulting composites, respectively, may vary according to the properties of the core and shell [16,17]. Shell may act as a protective material for the magnetic cores; at high temperatures, the absorption capabilities inherent in room temperature are retained [18]. Properties, such as interfacial polarization and corrosion resistance [19] are also improved. One can observe that shells may enhance the phase stability of cores at elevated temperatures, see [20,21,22] for example. Protective coatings may be applied, not only on individual particles, but also on microwires [23].

Of interest is the type of structure with a magnetic core and a dielectric shell. The core can be any magnetic element, ferrite alloy, or ceramic. Such dielectrics as Al_2_O_3_ [24], TiO_2_ [25], ZrO_2_ [26], SiO_2_ [27], carbon materials, and polymers are suitable as a dielectric shell [28,29]. Composites with a magnetic core and a dielectric shell are in increasing demand for technical applications, since the protective dielectric shell prevents the oxidation of the core and the agglomeration of the particles. In addition, the above type of structures combines dielectric and magnetic properties and acts as a mediator between dielectric and magnetic losses, with the possibility of modifying them by changing the synthesis conditions [30,31]. Composites of the “magnetic core-dielectric shell” type demonstrate better impedance matching and better absorbing properties when compared with other types of similar structures [32,33,34,35] in the microwave range (1–20 GHz) and can be considered as high-quality EM absorbers. Fe-SiO_2_ nano and microparticles can be used as a magnetism/pressure dual-mode sensor due to its magnetorheological properties [36,37], Fe-SiO_2_ coatings are applied in optical fiber-based corrosion sensors [38] where Fe-SiO_2_ particles used as corrosion proxy material are integrated into the optical fiber. Metal nanoparticles offer excellent localized surface plasmon resonance properties which exhibit a strong and well-defined color and enable visualization of color change [39]. Visual detection is based on metal–ligand coordination, where the metal and ligand act as electronic acceptor and donor, respectively. Based on this, Fe-SiO_2_ particles can be applied for recoverable the colorimetric sensor for Co^2+^ion [40]. Many studies of core-shell particles usually focus on chemical properties and methods to obtain such structures and usually overlook electromagnetic properties.

This research focuses on the synthesis and analysis of the properties of Fe-SiO_2_ core-shell powders. Carbonyl iron (CI) particles with a mean diameter of 2 μm were covered with a thick SiO_2_ dielectric shell using a modified sol–gel Stöber process [41]. The number of publications related to micro-sized iron particles and thick (>100 nm) SiO_2_ shells is next to none, despite the fact that the topic is relevant in the scientific community. It has been shown previously, that it was not possible to obtain shells with a thickness of more than 200 nm either by varying the duration of the deposition reaction or by changing the concentrations of the reagents [42]. In the present work, a technique is shown to obtain thick SiO_2_ shells of up to 475 nm as a result of a multi-iterative deposition of SiO_2_ on iron particles. Magnetostatic, electrodynamic, and thermal properties of the obtained samples are also studied.

## 2. Materials and Methods

### 2.1. Synthesis of Fe-SiO_2_ Powders

The deposition of the SiO_2_ shell onto particles of carbonyl iron (CI) was carried out using the modified Stöber process [41] (Equation (1)). Pure carbonyl iron powder (≥97.0 mass.% Fe), tetraethyl orthosilicate (TEOS) (CAS 78-10-4, Aldrich No. 86578), and ammonia solution (25%, reagent grade) were used. First, 1 g of the metal powder was immersed in ethanol (45 mL) in a round bottom 100-mL flask. The (TEOS) tetraethyl orthosilicate (4.5 mL) was added and ultrasonicated for 30 min, then, ammonia (3 mL) was added to the mixture.
*n*(SiOEt)_4_ + *2n*H_2_O + NH_3_ = *n*SiO_2_ + *4n*EtOH(1)

During the reaction, silicon oxide was synthesized in the form of individual particles and was deposited on carbonyl iron.

Since the maximum thickness of the shell is limited to 200 nm [42] in the one-stage Stöber synthesis, a multi-iterative deposition technique was studied. One “iteration” here was the full cycle of deposition including TEOS hydrolysis and air drying. Each “iteration” increased the thickness of the shell on the same particles already covered with SiO_2_. To simplify, the first iteration transformed Fe→“Fe-SiO_2_”; the second iteration gave “(Fe-SiO_2_)-SiO_2_”, and the third iteration resulted in a “((Fe-SiO_2_)-SiO_2_)-SiO_2_” product. In this study, pure carbonyl iron was studied and compared with powders, and underwent 1, 2, and 3 iterations.

The duration of the reaction was 2 h in each case, and the ratio [TEOS]/[NH_3_·H_2_O] = 1.5 remained constant. Between iterations, the product was air-dried at 60 °C 2 h, and, therefore, new layers of SiO_2_ were deposited on top of the already existing dielectric layers. At the end of the process, the main product, ferromagnetic powder, was isolated from the solution by decantation using a magnet. Following cleaning with ethanol, the powder was air-dried at 60 °C for 2 h.

### 2.2. Scanning Electron Microscopy

The morphology and elemental composition (energy-dispersive X-ray analysis, EDX) of the powders were analyzed using a JEOL JCM-7000 (JEOL, Tokyo, Japan) scanning electron microscope. The powders were dispersed on a conductive carbon adhesive tape, placed on an aluminum stand (no unwanted iron contorted the EDX data) and installed in a microscope chamber. A sample area of 140 × 100 µm was studied for the EDX analysis. More than 800 individual Fe-SiO_2_ particles were examined manually to study the particle size distribution and the shell thickness. The depth of the local X-ray spectral analysis was less than 5 µm. Iron particles covered the adhesive tape in a thick and dense layer to neglect the errors during the EDX analysis.

### 2.3. Vibrating-Sample Magnetometer (VSM)

A vibrating-sample magnetometer was used to study the magnetostatic parameters of iron powders. Fe-SiO_2_ powders were mixed with paraffin wax in a crucible with 20 vol.% of filler. First, the powder with molten paraffin was blended until the composites were completely mixed and hardened, and then a thin (0.5 × 7 mm) disk was formed from the resulting material, and the disk was placed in a magnetometer on a special holder.

### 2.4. Thermal Analysis

The thermal stability of the metal powders, as well as the impact of the protective shell of SiO_2_ on the oxidation of iron particles, were studied using the Netzsch STA 449 F3 Jupiter synchronous thermal analysis (STA) instrument (Netzsch, Germany). The samples were examined in the temperature range from 30 to 1000 °C with a heating rate of 10 degrees per minute. The measurements were carried out in air flow.

### 2.5. Microwave Measurements

For the microwave measurements, the composites were prepared as follows: Fe-SiO_2_ particles were mixed with paraffin, heated, and stirred until a homogeneous state. The filler volume in all cases was estimated as 20 vol.%. The samples were formed in the toroidal shape and placed inside a standard 7/3 mm coaxial transmission line [42]. S-parameters of the composite samples placed in the airline were measured in the frequency range of 0.1 to 20 GHz with a vector network analyzer (VNA). Ports at the end of the feeding coaxial cables were calibrated with standard TRL (Thru-Reflect-Line) calibration procedure with planes of phase reference at the ends of the measurement transmission line [43]. The complex microwave permeability and permittivity were determined with the standard Nicolson–Ross–Weir (NRW) [44,45] technique. The lower frequency boundary of 0.1 GHz was determined by the sensitivity of the measurement system. Half-wavelength resonances and higher-order modes arose at the upper boundary of 20 GHz [46,47]. These resonances are not related to the material parameters of the composites, but noticeably affect the NRW solution.

## 3. Results and Discussion

### 3.1. Theoretical Calculation of the Effective Permittivity

The effective permittivity of the composites was calculated using the Maxwell Garnett formula [48], since this formula is most suitable for describing the ‘metal sphere with a dielectric shell in a paraffin matrix model without the formation of clusters [49,50]. In the calculation, the thickness of the dielectric shell was varied from 30 to 550 nm. Since the formula describes two-component mixtures, the calculation was carried out in two stages. At the first stage, the permittivity of the iron particles + shell-matrix SiO_2_, (Fe-SiO_2_)’ was calculated. In this case, the volume fractions of the Fe and SiO_2_ components were calculated from geometric considerations based on the data from magnetostatic measurements, elemental analysis results, and images from an electron microscope. Effective permittivity of the composites in the quasi-static field was calculated as follows:(2)$$\varepsilon_{eff}=\varepsilon_{h}+3f\varepsilon_{h}\frac{\varepsilon_{i}-\varepsilon_{h}}{\varepsilon_{i}+2\varepsilon_{h}-f\left(\varepsilon_{i}-\varepsilon_{h}\right)}$$
where ε*_h_* and ε*_i_* were the relative permittivity of the matrix and inclusion, respectively, and *f* was a volume fraction of the inclusion. At the second stage, the obtained values were used to calculate the properties of the ‘Fe-SiO_2_ + paraffin matrix, (Fe-SiO_2_ + Wax)’ composite. The calculations used ε(Fe) = ∞, ε(Paraffin) = 2.4, and ε(SiO_2_) = 4 [51], and the volume fraction of the filler Fe + SiO_2_ in the paraffin matrix was 20%. As the shell thickness increased from 30 to 550 nm, the effective permittivity of the Fe-SiO_2_ structure decreased from 200 to 10; by a factor of 20. However, with a further increase in the shell thickness, the efficiency of reducing the dielectric constant decreased. In this case, it is not important how thick (up to 550 nm in the theoretical model) dielectric shells are obtained, since only the geometric configuration of the core-shell structure is important.

The effective permittivity of the final composite in the paraffin matrix decreased by 20% with an increase in the shell thickness from 30 to 550 nm (Figure 1a,b). Considering all of the above, we can conclude that a thick SiO_2_ dielectric shell, in theory, can significantly reduce the effective permittivity of the composite, which greatly improves the impedance matching and, consequently, the absorbing properties of the material.

### 3.2. Scanning Electron Microscopy Analysis

With an increase in the number of iterations of applying the SiO_2_ shell from one to three, an increase in the thickness of the shell from ≈150 nm to 475 nm was observed on individual particles (Figure 2, Table 1). During the hydrolysis reaction, SiO_2_ is formed both in the form of individual particles (Figure 3) and in the form deposited on the surface of iron particles. Most of the individual particles are disposed of during the cleaning process. A detailed quantitative change in the particle size can be observed in the histogram (Figure 4) obtained from the analysis of the electron microscopy images, using Toupview software 3.7 version.

From the data presented on the histogram and the cumulative curve (Figure 5), one can observe that the maxima in the particle size distribution shifts to the right when the next iteration is added in the process of applying the SiO_2_ shell. In the range of the boundary values, the particles that are not typical of their size can be encountered, due to the small sample size (200 particles of each size), and the initial heterogeneity of carbonyl iron particles in size. However, it is easy to see that the maximum in the distribution of the particles obtained in the three-iteration process is shifted to the right by 1 µm, compared to uncoated particles. If we take into account that the original iron has the same size distribution for all three experiments, then the additional size falls exclusively on the SiO_2_ dielectric shell, which is exactly 450–500 nm and consistent with the data obtained from the analysis of the images from an electron microscope.

Considering all of the above, we can conclude that when applying the SiO_2_ dielectric shell repeatedly, with intermediate air drying of the powder between iterations, it is possible to obtain a thick, up to 475 nm, SiO_2_ shell. It shows that not only SiO_2_ deposits onto the iron surface, but also it is formed on the SiO_2_ surface as well. It is important to note that it was not possible to obtain dielectric shells with a thickness of 475 nm in one stage of the modified Stöber process. At a reaction time of 6 h, with the constant addition of TEOS and NH_4_OH (to avoid complete consumption of the reagents), the thickness of the deposited SiO_2_ shell did not exceed 200 nm.

### 3.3. Elemental Analysis

According to the EDX analysis, with an increase in the number of iterations from one to three, the SiO_x_ ratio decreased from *x* = 3.8 to 2.8, respectively. (Figure 6a,b). The ratio of iron to silicon also decreased from 7.8 to 2.5 with an increase in the number of iterations from one to three, which indicated a decrease in the iron fraction and an increase in the silicon fraction, i.e., an increase in the thickness of the SiO_2_ dielectric shell.

### 3.4. Magneto Static (Vibrating Scanning Magnetometer Measurements)

When analyzing the magnetization curves, it can be seen that for all three series of experiments, the saturation field is 5–6 kOe (Figure 7). The saturation magnetization in this series of experiments showed the systematics: With an increase in the number of iterations of SiO_2_ deposition on iron particles, the saturation magnetization of the composites (Fe-SiO_2_ + paraffin) decreased from 230 to 175 G, for one and three iterations, respectively (Figure 8). The decrease in the saturation magnetization is associated with a decrease in the magnetic (iron) fraction in the composites, and, consequently, with an increase in the SiO_2_ content and an increase in the thickness of the dielectric shell.

To calculate the thickness of the dielectric layer on iron particles, we need to determine the volume concentration of the pure iron in the resulting composite. For that purpose, the VSM was used with the following procedure:

Composites consist of iron, SiO_2_ shells, paraffin, and air pores (for such composites was found to be around 5%). The thickness of iron oxide formed on the surface of iron particles is no more than 3 nm, and the contribution of this effect can be neglected in comparison with other errors that occur during measurements [19,52]. The saturation magnetization of pure iron is known as 1700 Gauss; therefore, the volume fraction of iron can be calculated from the magnetization of the composites. Knowing the volume and mass fractions of iron and paraffin, one can determine the volume and mass fractions of SiO_2_. Knowing the volume fractions of SiO_2_, and assuming that all SiO_2_ are deposited in the form of shells on spherical iron particles with an average size of 1.5 μm, taking into account the geometry, it is possible to calculate the thickness of the SiO_2_ shell. The thicknesses of the dielectric shells calculated in this way are slightly larger than the thicknesses obtained from the SEM image analysis, probably because iron particles are not the same size and some part of SiO_2_ exists in the form of individual particles, which our model cannot take into account.

The shell thicknesses calculated from the data of the magnetostatic measurements are in good agreement with the thicknesses obtained by processing images of scanning electron microscopy. Thickness values calculated from the magnetostatic measurements are slightly higher (from 20 to 100 nm higher) than those obtained from SEM images (Figure 9). We can conclude that the calculation of thicknesses from the data of the magnetostatic measurements is consistent with the real thicknesses obtained as a result of SEM image processing. With further modification of the calculation methods, namely, taking into account the volume of air inclusions (or completely eliminating air pores in composites by means of high pressure and high pressing density) and taking into account the size dispersion of carbonyl iron particles, it will be possible to evaluate the morphology and structural features of the powders, using magnetic measurements and theoretical calculations based on mixing formulas.

### 3.5. Synchronous Thermal Analysis (TGA & DSC)

The results of the oxidization of the prepared materials utilizing synchronous thermal analysis are shown in Figure 10. The reaction mechanism of iron oxidation to the highest oxide is complex and occurs with the production of short-lifetime intermediate phases FeO-Fe_3_O_4_-Fe_2_O_3_ [17,53]. Oxidation of the CI produces a single exothermic DSC-maxima with piecemeal weight increasing to 38.79% (Figure 10A). Oxidation of uncoated iron particles at linear heating demonstrates a single exothermic reaction peak with piecemeal weight increasing to 38.79% (see Figure 10A).

Protective coatings based on silica prevent direct contact between material and the oxidizing environment, display an improved thermal stability behavior that does not depend on the choice of the source precursor of Si [22]. For instance, the onset temperature of lithium-rich layered oxides coated with SiO_2_ utilizing a sol–gel process shifts from 217.84 for uncoated material to 245.78 °C [54]. Furthermore, nanocomposite AlCr(Si)N protective coatings with increasing Si-content protect against oxidization [55]. The addition of the coating on a substrate surface increases the onset of oxidation from 1100 up to 1260 °C and protects from further oxidation.

Herein, the presence of SiO_2_-coating on the carbonyl iron particles prevents the diffusion of oxygen oxidation from the coating surface to the iron core during the early stages of oxidation [54,55]. That displacement of the peak toward the higher temperatures (see Figure 10B–D), enhances the thermal stability by 5% and, also, the hydrophobicity [56] and corrosion resistance [19]. The increase in the thickness of the SiO_2_ shell demonstrates a shift of oxidation onset temperature to the right from 166.7 (for pure CI) to 331.2 °C (for a thick 450 nm shell coating)—which is shown with big dots in Figure 10. The last two peaks shift to the high-temperature region (Figure 10B–D). The value of residual energy reduces as the volume fraction of silica increases from 6 kJ/g to 3.8 kJ/g. The mass gain of core-shell particles occurred in four steps accompanied by exothermic peaks. The mass change of the first, second and fourth peaks (Figure 10B–D) of coated powders are numerically equal to the mass gain of CI oxidization. The observed “split” of the single oxidization peak of CI, probably, indicates the separation of intermediate phases at the iron oxidization by means of SiO_2_-coating. The third DSC-maximum characterized by plateau and a slight increase in the TG-curve (especially for the one and two iterative process, see Figure 11) may be descriptive of the reaction between the core and shell in the 620–750 °C range. The phase constitution of “split” maxima can be studied after annealing to a convenient temperature for each exothermic peak in a furnace in a dry air atmosphere by means of XRD analysis in further research.

### 3.6. Microwave Permittivity and Permeability

When studying the frequency dispersions of the permittivity and permeability of composites for different numbers of iterations of SiO_2_ deposition to carbonyl iron particles, certain systematics are observed. The slope of the curve of the real part of the permittivity and permeability (Figure 12A) varies, depending on the number of iterations. The value of ε′ gradually decreases with increasing frequency for all three samples. We see a noticeable decrease of the effective permittivity of the composite with the increase of shell thickness. We can conclude that the effective permittivity of the composite is determined mainly by spatial separation of the particles in the composite dielectric matrix, and consequently by the effective electrical capacitance between adjacent particles. Errors of measured permittivity and permeability at frequencies below 1 GHz are attributed to a lower sensitivity of the transmission/reflection method due to a small optical thickness of the studied samples. Resonance at frequency ~16 GHz may be attributed to the emergence of higher-order modes on a sample boundary [46]. This resonance cannot be attributed to a half-wavelength resonance, since a half-wavelength resonance on the sample is estimated to be higher in frequency (approx. 38 GHz).

From the Maxwell Garnett approximation, the quasi-static values of ε′ were expected to decrease with an increase in the number of iterations, since the thickness of the dielectric shell increased (Figure 1). Experimental data are consistent with theoretical predictions—powders with thicker SIO_2_ shells correspond to lower values of the static dielectric constant, the theoretical model predicted a decrease in ε′ by 16% with the increasing shell thickness from 100 to 500 nm, in the experiment the decrease was 8–10%. The absolute values obtained in the experiment differ by less than 20% from the theoretical approximations. Larger dielectric losses correspond to composites with a larger shell, but the ε″ values are relatively small (0.1–0.2) that are comparable to the measurement errors due to the VNA drift and waveguide setup instability.

The real and imaginary parts of the magnetic permeability also decrease with an increase in the number of iterations (Figure 12C,D), which confirms the results of the magnetostatic measurements. Unlike ε′, with an increase in the number of iterations, only the absolute values of μ′ and μ″ decrease, but the frequency dispersion remains constant for all three cases. This suggests that the magnetic properties of the resulting powders do not deteriorate when a thicker shell is applied. The frequency dependences of μ′ at different thicknesses of SiO_2_ differ by a constant factor, which means that the mixing formula that describes the behavior of these composites near or below a given concentration is Wiener’s law. Wiener’s mixing rule says that μ′ is proportional to the concentration of the magnetic phase—the more iron in the composite, the higher the value of the magnetic permeability. In other words, there is no significant interaction between particles.

## 4. Conclusions

Hydrolysis of TEOS in the presence of ammonia gave a uniform SiO_2_ coating on a surface of carbonyl iron micro-particles. Using the proposed method of multi-iterative deposition of SiO_2_ with air drying between iterations, it was possible to obtain thick (up to 475 nm) SiO_2_ protective shells on micron-sized carbonyl iron particles. The presence of protective shells was confirmed by three independent methods: SEM, VSM, and the matching between microwave measurement data with the theoretical model. The 475 nm SiO_2_ shell leads to a decrease in the permittivity and permeability of the Fe-SiO_2_-wax composites in the microwave frequency range. The decrease in the permittivity and permeability is associated with a quantitative decrease in the magnetic (iron) fraction in the medium of the composite and corresponds to the Maxwell Garnett medium theory. The frequency dependences of μ′ at different thicknesses of SiO_2_ differ by a constant factor, μ′ is proportional to the concentration of the iron and there is no significant interaction between particles. Thicker protective shells improved the thermal protection of the iron particles by shifting the transition temperature to the higher oxide from 600 to 900 degrees Celsius and shifting the oxidation onset temperature from 166 to 331 °C.

All of the above allow us to conclude that Fe-SiO_2_ particles with a thick dielectric shell, obtained by the multi-iteration method, improve the thermal protection of iron particles, prevent corrosion, enhance, and modify the electrodynamic parameters of the composites. These functional materials can be used in many biomedical, magneto physical and engineering applications where thermal stability in combination with excellent fine-tuned electromagnetic properties is required.

## Figures and Tables

**Figure 1 sensors-23-01727-f001:**
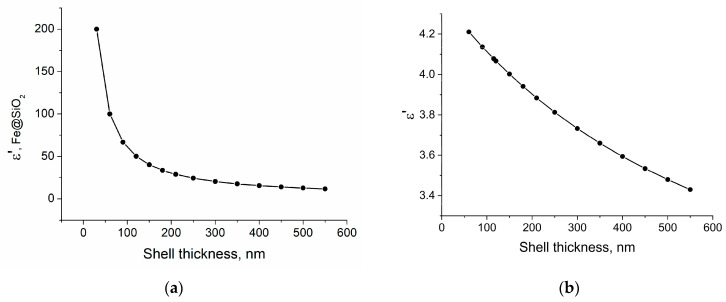
(**a**) Dependence of ε eff (Fe-SiO_2_) on the SiO_2_ shell thickness. (**b**). Dependence of ε eff (Fe-SiO_2_ + wax) on the SiO_2_ shell thickness.

**Figure 2 sensors-23-01727-f002:**
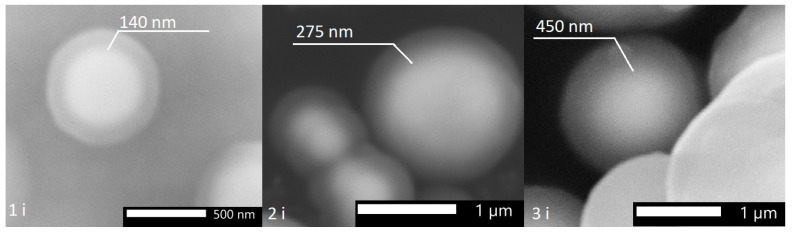
SEM images of Fe-SiO_2_ particles with a different number of iterations.

**Figure 3 sensors-23-01727-f003:**
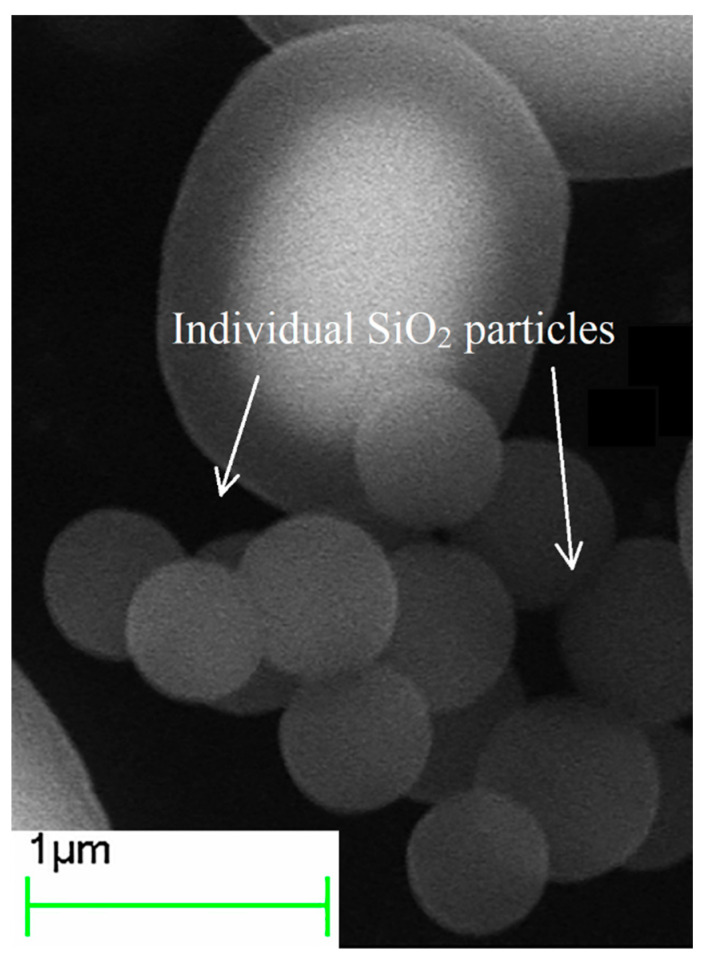
Images of individual SiO_2_ particles.

**Figure 4 sensors-23-01727-f004:**
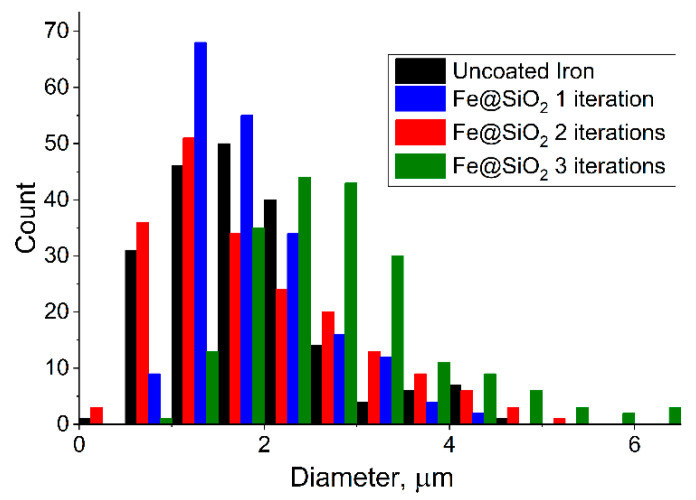
Particle size distribution measured for pure carbonyl iron,; one, two, and three iterations.

**Figure 5 sensors-23-01727-f005:**
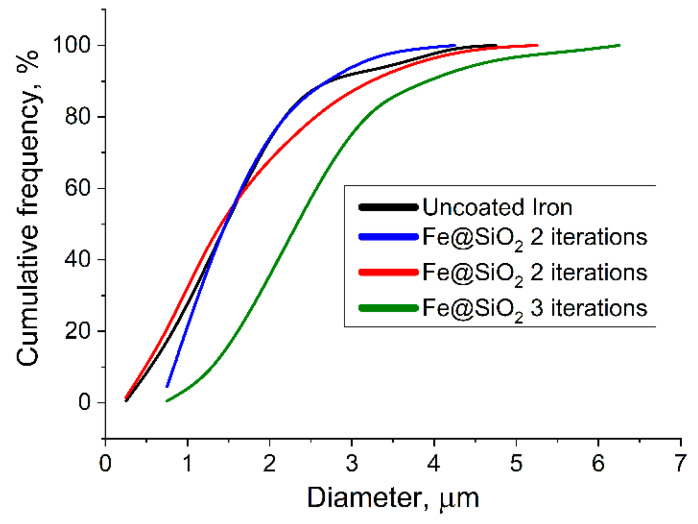
Cumulative particle size curve for uncoated CI; one, two, and three iterations.

**Figure 6 sensors-23-01727-f006:**
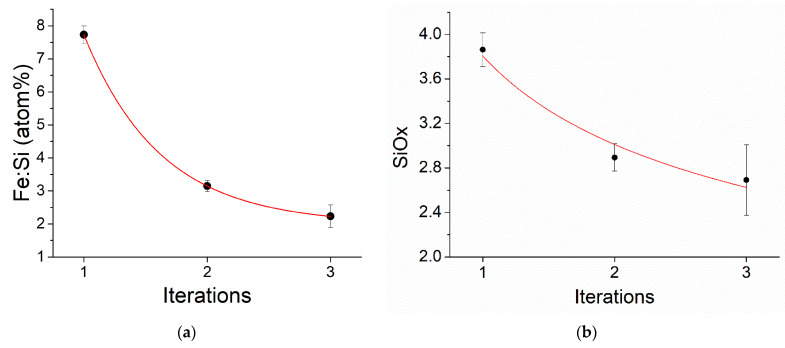
(**a**) Fe:Si ratio and (**b**) Si:O ratio with a variation in the number of iterations.

**Figure 7 sensors-23-01727-f007:**
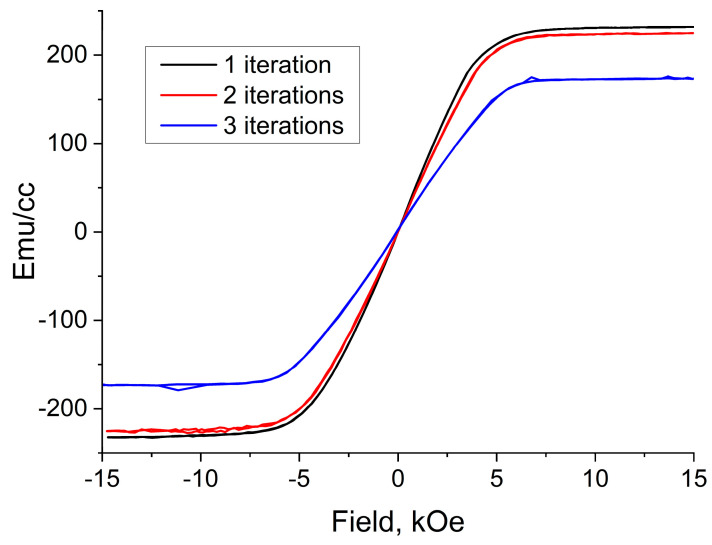
Magnetization curves in an iterative series of experiments.

**Figure 8 sensors-23-01727-f008:**
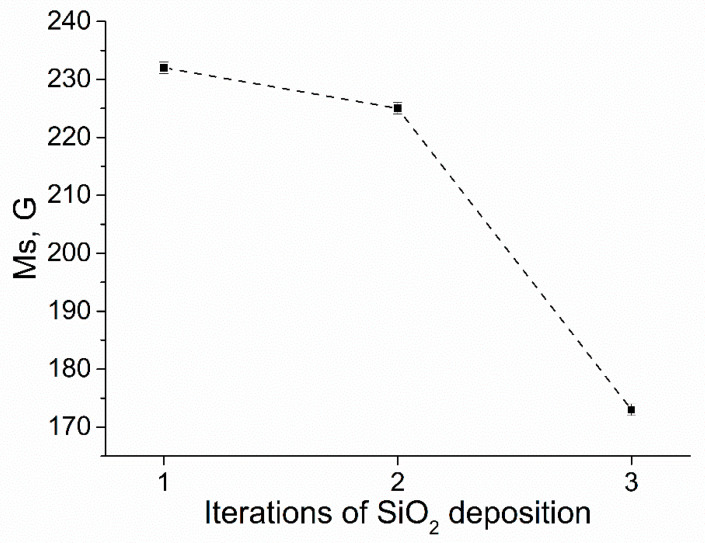
Saturation magnetization for one, two, and three iterations of deposition of the dielectric shell.

**Figure 9 sensors-23-01727-f009:**
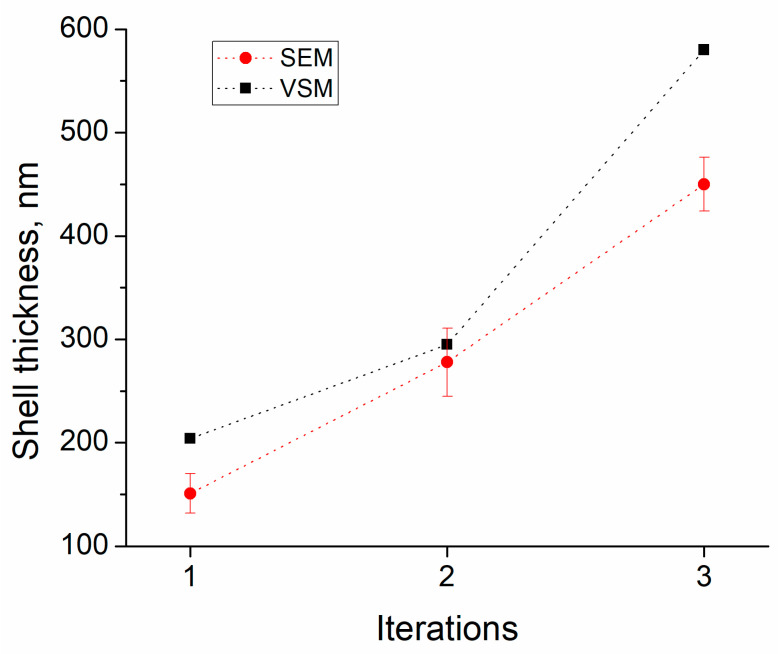
Comparison of the SEM and VSM methods.

**Figure 10 sensors-23-01727-f010:**
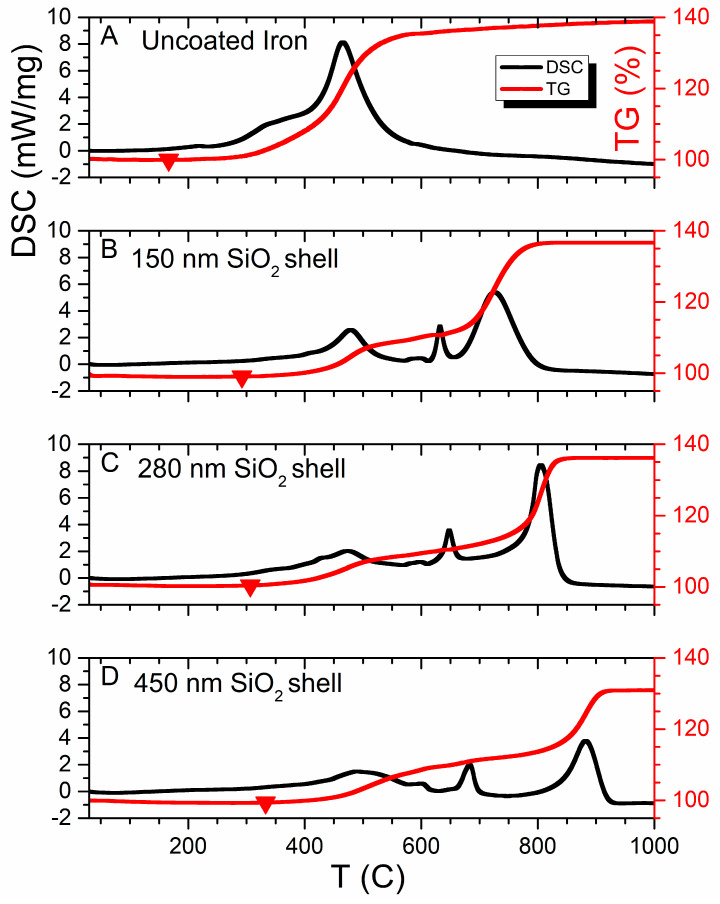
Synchronous thermal analysis results. Comparison of the curves for pure iron (**A**) and Fe-SiO_2_ particles in cases of one, (**B**) two (**C**) and three (**D**) iterations.

**Figure 11 sensors-23-01727-f011:**
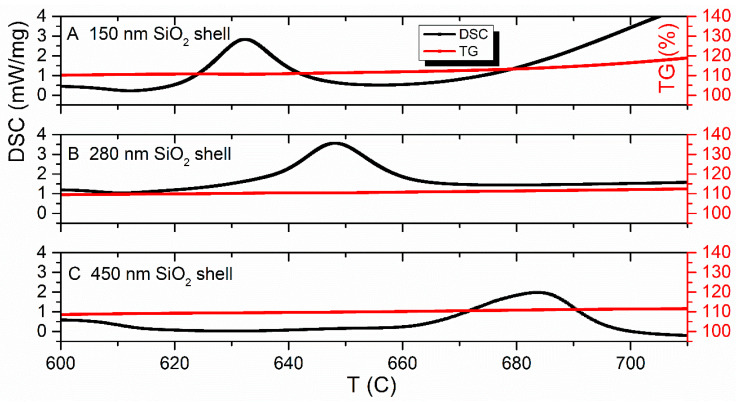
Synchronous thermal analysis results. Detailed 600–720 C temperature interval.

**Figure 12 sensors-23-01727-f012:**
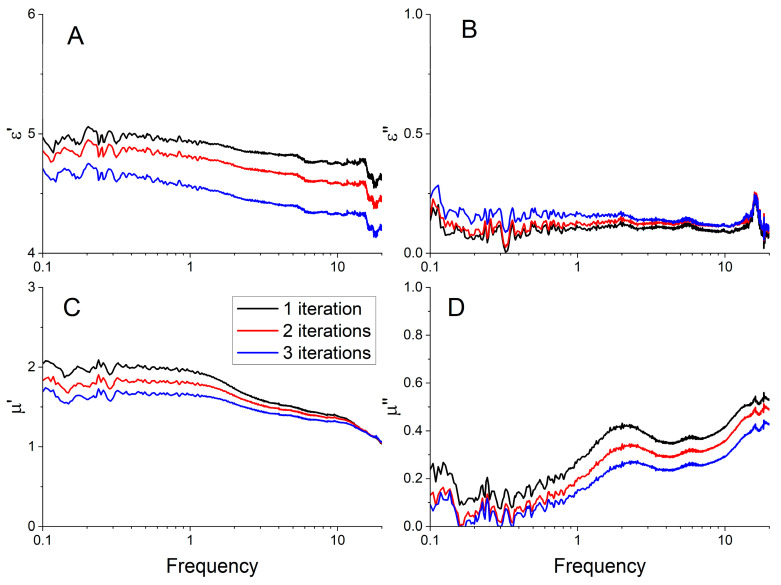
Frequency dispersion of permittivity (**A**,**B**) and permeability (**C**,**D**) in the experiments with a different number of iterations of the SiO_2_ shell deposition.

**Table 1 sensors-23-01727-t001:** SiO_2_ shell thicknesses vs. the number of iterations of SiO_2_ deposition on iron particles.

Number of Iterations	1	2	3
Thickness of SiO_2_	151 ± 19 nm	278 ± 30 nm	450 ± 26 nm

## Data Availability

Data is contained within the article.

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
