# Peer review of "Deposition of Thick SiO2 Coatings to Carbonyl Iron Microparticles for Thermal Stability and Microwave Performance"

_sensors, 2023, doi:10.3390/s23031727_

Round 1

Reviewer 1 Report

The subject of the present study is very interesting and might be suitable for publication. The manuscript is submitted to the MDPI Sensors journal. Although the selected topic “Synthesis and Applications of Nanostructured Metals and Metal Oxides” is correct, the whole manuscript development of the subject might better focused on the suitability of the proposed materials for the applications as components of microwave devices, magnetic sensors or simply as a protecting coverings of biomedical components (see examples: Chiriac et al.; Microwire array for giant magnetoimpedance detection of magnetic particles for biosensor prototype. J. Magn. Magn. Mater. 2007, 311, 425–428 Safronov, et al.  Polyacrylamide ferrogels with magnetite or strontium hexaferrite: Next step in the development of soft biomimetic matter for biosensor applications. Sensors 2018, 18, 257; Darton, N.J.; Ionescu, A.; Llandro, J. Magnetic Nanoparticles in Biosensing and Medicine; Cambridge University Press: Cambridge, UK, 2019; p. 279 and others).

The title should be re-thought as it does not explain the type of the material under consideration (microparticles) and quite popular but not accurate expression “electrodynamic performance” should be also clarified – what exactly was studied and understood?

Yes, there is a tradition to call these particles in the initial state as “pure iron”, however it is most likely they are core/shell type particles covered by the iron oxide and therefore the whole story become more complex as the FeOx/SiO2 shell may contribute to the electrodynamics in a different way.

As the microwave studies are very important, at least brief description of the main point must be provided apart from ref. 38, which does not give a complete feeling of the device also.

The concept of iterations is given in very confusing way – the size distribution should be given for 800 particles of each type. The word “iterations” is confusing, the size cannot be changed by the repetition of measurements for which they typically use concept of iterations.

Synchronous thermal analysis results are very interesting but the reason for the peak displacement toward the higher temperatures requires additional explanation.

All figure must either show error bars or contain some kind of the experimental accuracy evaluation in the main text (see figure 8).

Work is very interesting but it requires some polishing, style correction and better description to be useful for the other researches and multidisciplinary audience of Sensors journal.

Author Response

Please see the attachment.  The attached file contains answers for all three reviewers in "point-by-point" style.

Reviewer 2 Report

The paper titled “Deposition of thick SiO2 coatings to carbonyl iron for thermal stability and electrodynamic performance” was aimed on the synthesis and analysis of the magnetostatic, electrodynamic, and thermal properties of Fe@SiO2 core-shell particles with a thickness of SiO2 shells up to 500 nm, which was achievable due to multi-iterative deposition of SiO2 on iron particles. The paper is well-written and can be recommended for publication after minor revision. The comments are listed below.

1.     All the abbreviations should be defined when first used. For instance, “DC arc plasma” should be “direct current (DC) arc plasma”

2.     Regarding the application of the core-shell approach, Prof. Volodin has reported a number of papers (Zaikovskii et al. Effect of Carbon Coating on Spontaneous C12A7 Whisker Formation. Applied Surface Science. 2018. V.444. P.336-338. DOI: 10.1016/j.apsusc.2018.03.056; Yakovlev et al. Stabilizing Effect of the Carbon Shell on Phase Transformation of the Nanocrystalline Alumina Particles. Ceramics International. 2018. V.44. N5. P.4801-4806. DOI: 10.1016/j.ceramint.2017.12.066; Bedilo et al. Silica-Coated Nanocrystalline TiO2 with Improved Thermal Stability. Ceramics International. 2019. V.45. N2. P.3547-3553. DOI: 10.1016/j.ceramint.2018.11.013; Stoyanovskii et al. Effect of Carbon Shell on Stabilization of Single-Phase Lanthanum and Praseodymium Hexaaluminates Prepared by a Modified Pechini Method. Ceramics International. 2020. V.46. N18, Part A. P.29150-29159. DOI: 10.1016/j.ceramint.2020.08.088). In these works, various shells were applied to enhance the phase stability at elevated temperatures. I guess, mentioning of these works can improve the Introduction of the present paper.

3.     Line 90: “Co2+ion” -> “Co2+ ion”

4.     Lines 94-95: “Carbonyl iron (CI) powder with a mean diameter of 2 μm with a thick SiO2 dielectric shell, deposited on CI with a sol-gel modified Stober process [37].” -> “Carbonyl iron (CI) particles with a mean diameter of 2 μm were covered with a thick SiO2 dielectric shell using a modified sol-gel Stöber process [37].”

5.     Line 116: “TEOS” should be defined.

6.     Line 138: “VSM” should be defined.

7.     Lines 168-170: “was calculated as:” -> “was calculated as follows:”, then the equation should come, and finally, the description in the brackets. The number of the equation is incorrect (another equation in Lines 111-112 is already numbered as 1).

8.     Lines 175-176: “However, with a further increase in the shell” -?-> thickness?

9.     It is not clear from the description of Figure 1, is it one iteration Fe@SiO2 sample or not.

10.  The highest shell thickness is rounded to 500 nm, which is incorrect. As it follows from Table 1, the maximum value does not exceed 476 nm. This should be corrected all over the text.

11.  Figures 4 and 5 can be combined in one. Then, “mkm” in the X-axis title should be corrected as “µm” or “micron”.

12.  All over the text: “Stober” -> “Stöber”

13.  All over the text: “hour”, “hours” -> “h”

14.  Line 223: “the SiOx ratio” – What is SiOx? What is the structure of such oxide? Is it possible that iron oxide is formed under the synthesis conditions?

15.  If iron oxide is formed, can this affect the shell thickness values calculated from the data of the magnetostatic measurements?

16.  Figure 10, caption: Comma is missing.

17.  Line 227: No brackets are required.

18.  Line 334: The last sentence looks like incomplete one.

19.  Lines 353: degrees Celsius” –> “°C”.

Author Response

(The authors gave the same response as above.)

Reviewer 3 Report

The paper shows that thick SiO2 protective layer reduces the permittivity of the ‘Fe@SiO2 - paraffin’ composite in accordance with the Maxwell Garnett me-dium theory. This work is interesting for the reader in field of semiconductivity, after answering the following questions, it can be published in sensors:

1) Fig. 10, the dsc curves are not in the same scale, and difficult to be compared. Fig. 11 also has the same question.

2) SEM shows the shell of siO2 around the Fe core, and give the thickness. Is it proved by XRD or XPS analysis?

3) Fig. 12, generally, the permeability of the material is correlated with the Ms, and with changing iteration number, they has not a consistent variation tendency, please give some comments.,

4) Fig. 4, the unit of diametr is wrong.

5) The reference format can be further unified.

Author Response

(The authors gave the same response as above.)

Round 2

Reviewer 1 Report

Work can be published in the present state.